# Analog quantum simulation of the Rabi model in the ultra-strong coupling regime

Jochen Braumüller[1], Michael Marthaler[2], Andre Schneider[1], Alexander Stehli[1], Hannes Rotzinger[1], Martin Weides [1,3] & Alexey V. Ustinov[1,4]

The quantum Rabi model describes the fundamental mechanism of light-matter interaction. It consists of a two-level atom or qubit coupled to a quantized harmonic mode via a transversal interaction. In the weak coupling regime, it reduces to the well-known Jaynes–Cummings model by applying a rotating wave approximation. The rotating wave approximation breaks down in the ultra-strong coupling regime, where the effective coupling strength $g$ is comparable to the energy $\omega$ of the bosonic mode, and remarkable features in the system dynamics are revealed. Here we demonstrate an analog quantum simulation of an effective quantum Rabi model in the ultra-strong coupling regime, achieving a relative coupling ratio of $g/\omega \sim 0.6$. The quantum hardware of the simulator is a superconducting circuit embedded in a cQED setup. We observe fast and periodic quantum state collapses and revivals of the initial qubit state, being the most distinct signature of the synthesized model.

[1] Physikalisches Institut, Karlsruhe Institute of Technology, Wolfgang-Gaede-Straße 1, 76131 Karlsruhe, Germany. [2] Institut für Theoretische Festkörperphysik, Karlsruhe Institute of Technology, Wolfgang-Gaede-Straße 1, 76131 Karlsruhe, Germany. [3] Physikalisches Institut, Johannes Gutenberg University Mainz, Staudinger Weg 7, 55128 Mainz, Germany. [4] Russian Quantum Center, National University of Science and Technology MISIS, Leninsky Ave 4, Moscow 119049, Russia. Correspondence and requests for materials should be addressed to J.Bül. (email: jochen.braumueller@kit.edu)

Finding solutions to many quantum problems is a very challenging task[1]. The reason is the exponentially large number of degrees of freedom in a quantum system, requiring computational power and memory that easily exceed the capabilities of present classical computers. A yet to be demonstrated universal digital quantum computer of sufficient size would be capable of efficiently solving most quantum problems[1, 2]. A more feasible approach to achieve a computational speedup in the near future is quantum simulation[1–3]. In the framework of analog quantum simulation, a tailored and well-controllable artificial quantum system is mapped onto a quantum problem of interest in order to mimic its dynamics. Since the same equations of motion hold for both systems, the solution of the underlying quantum problem is inferred by observing the time evolution of the artificially built model system, while making use of its intrinsic quantumness. This scheme may be applied to the simulation of complex quantum problems, in the spirit originally proposed by Feynman[1].

Quantum simulation was performed on various experimental platforms. Examples of analog quantum simulation are the study of fermionic transport[4] and magnetism[5] with cold atoms and the simulation of a quantum magnet and the Dirac equation with trapped ions[6, 7]. The exploration of non-equilibrium physics was proposed with an on-chip quantum simulator based on superconducting circuits[8, 9]. Digital simulation schemes with superconducting devices were demonstrated for fermionic models[10] and spin systems[11].

The quantum Rabi model in quantum optics describes the interaction between a two-level atom and a single quantized harmonic oscillator mode[12, 13]. In the weak coupling regime, which may still be strong in the sense of quantum electrodynamics (QED), a rotating wave approximation (RWA) can be applied and the Rabi model reduces to the Jaynes–Cummings model[14], which captures most relevant scenarios in cavity and circuit QED. In the ultra-strong coupling (USC) and deep strong coupling regimes, where the coupling strength is comparable to the mode energies[15], the counter rotating terms in the interaction Hamiltonian can no longer be neglected and the RWA breaks down. As a consequence, the total excitation number in the quantum Rabi model is not conserved. Except for one recent paradigm of finding an exact solution[16], an analytically closed solution of the quantum Rabi model does not exist due to the lack of a second conserved quantity which renders it non-integrable. The quantum Rabi model, in particular in the USC regime and beyond, exhibits non-classical features and rising interest in it is inspired by strong advances of experimental capabilities[15, 17–19]. The specific spectral features of the USC regime and the consequent breakdown of the RWA were previously observed with a superconducting circuit by implementing an increased physical coupling strength[20, 21]. A similar approach involving a flux qubit coupled to a single-mode resonator allowed to access the deep strong coupling regime in a closed system[22]. The USC regime was reached before by dynamically modulating the flux bias of a superconducting qubit, reaching a coupling strength of about 0.1 of the effective resonator frequency[23].

In our approach, we engineer an effective quantum Rabi Hamiltonian with an analog quantum simulation scheme based on the application of microwave Rabi drive tones. By a decrease of the subsystem energies, the USC condition is satisfied in the effective rotating frame, allowing to observe the distinct model dynamics. The scheme may be a route to efficiently generate non-classical cavity states[24–26] and may be extended to explore relevant physical models such as the Dirac equation in $(1 + 1)$ dimensions. Its characteristic dynamics is expected to display a Zitterbewegung in the spacial quadrature of the bosonic mode[27]. This dynamics has been observed with trapped ions[7], likewise

based on a Hamiltonian that is closely related to the USC Rabi model. It has been shown recently that a quantum phase transition, typically requiring a continuum of modes, can appear already in the quantum Rabi model under appropriate conditions[28]. The experimental challenge is projected to the coupling requirements in the model which may be accomplished with the simulation scheme presented. This can be a starting point to experimentally investigate critical phenomena in a small and well-controlled quantum system[29]. With a digital simulation approach, the dynamics of the quantum Rabi model in USC conditions was similarly studied very recently[30].

In our experiment we simulate the quantum Rabi model in the USC regime achieving a relative coupling strength of up to 0.6. Dependent on our experimental parameters, we observe periodically recurring quantum state collapses and revivals in the qubit dynamics, being a distinct signature of USC. The collapse-revival dynamics appears most clearly in the absence of the qubit energy term in the model, according to the expectation from master equation simulations. In addition, we use our device to simulate the full quantum Rabi model and are able to observe the onset of an additional substructure in the qubit time evolution. With this proof of principle experiment we validate the experimental feasibility of the analog quantum simulation scheme and demonstrate the potential of superconducting circuits for the field of quantum simulation.

## Results

**Simulation scheme.** The quantum Rabi Hamiltonian reads

$$\frac{\hat{H}}{\hbar} = \frac{\epsilon}{2}\hat{\sigma}_z + \omega\hat{b}^\dagger\hat{b} + g\hat{\sigma}_x\left(\hat{b}^\dagger + \hat{b}\right), \qquad (1)$$

with $\epsilon$ the qubit energy splitting, $\omega$ the bosonic mode frequency and $g$ the transversal coupling strength. $\hat{\sigma}_i$ are Pauli matrices with $\hat{\sigma}_z|g\rangle = -|g\rangle$ and $\hat{\sigma}_z|e\rangle = |e\rangle$, where $|g\rangle$, $|e\rangle$ denote eigenstates of the computational qubit basis. $\hat{b}^\dagger$ ($\hat{b}$) are creation (annihilation) operators in the Fock space of the bosonic mode. Both elements of the model are physically implemented in the experiment, with a small geometric coupling $g \ll \epsilon, \omega$, such that the RWA applies and Eq. (1) takes the form of the Jaynes–Cummings Hamiltonian. In order to access the USC regime, we follow the scheme proposed in ref. [27]. It is based on the application of two transversal microwave Rabi drive tones coupling to the qubit. The USC condition is created in a synthesized effective Hamiltonian in the frame rotating with the dominant drive frequency. In this engineered Hamiltonian, the effective mode energies are set by the Rabi drive parameters. The Jaynes–Cummings Hamiltonian in the laboratory frame with both drives applied takes the form

$$\begin{aligned}\frac{\hat{H}_d}{\hbar} =\ & \frac{\epsilon}{2}\hat{\sigma}_z + \omega\hat{b}^\dagger\hat{b} + g\left(\hat{\sigma}_-\hat{b}^\dagger + \hat{\sigma}_+\hat{b}\right) \\ & + \hat{\sigma}_x\eta_1\cos(\omega_1 t + \varphi_1) + \hat{\sigma}_x\eta_2\cos(\omega_2 t + \varphi_2),\end{aligned} \qquad (2)$$

with $\eta_i$ the amplitudes and $\omega_i$ the frequencies of drive $i$. $\varphi_i$ denotes the relative phase of drive $i$ in the coordinate system of the qubit Bloch sphere in the laboratory frame. Within the RWA where $\eta_i/\omega_i \ll 1$, the $\varphi_i$ enter as relative phases of the transversal coupling operators $e^{-i\varphi_i}\hat{\sigma}_+ + \text{h.c.}$, where $\hat{\sigma}_\pm = 1/2\left(\hat{\sigma}_x \pm i\hat{\sigma}_y\right)$ denote Pauli's ladder operators. In the following, we set $\varphi_i = 0$ to recover the familiar $\hat{\sigma}_x$ coupling without loss of generality. Going to the frame rotating with $\omega_1$ and neglecting terms rotating with $e^{\pm 2i\omega_1 t}$ renders the first driving term time-independent, yielding

$$\begin{aligned}\frac{\hat{H}_1}{\hbar} =\ & (\epsilon - \omega_1)\frac{\hat{\sigma}_z}{2} + (\omega - \omega_1)\hat{b}^\dagger\hat{b} + g\left(\hat{\sigma}_-\hat{b}^\dagger + \hat{\sigma}_+\hat{b}\right) \\ & + \frac{\eta_1}{2}\hat{\sigma}_x + \frac{\eta_2}{2}\left(\hat{\sigma}_+ e^{i(\omega_1 - \omega_2)t} + \hat{\sigma}_- e^{-i(\omega_1 - \omega_2)t}\right).\end{aligned} \qquad (3)$$

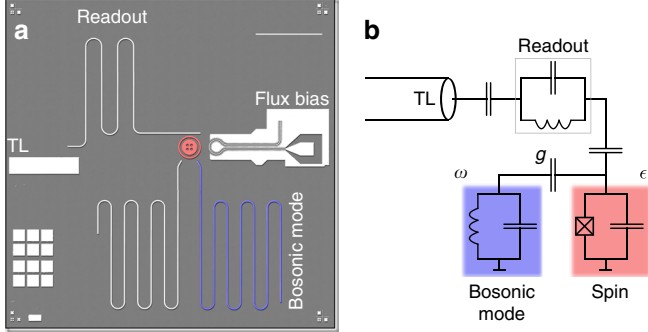

**Fig. 1** Quantum simulation device. **a** Optical micrograph with the atomic spin represented by a concentric transmon qubit, highlighted in *red* and the $\lambda/2$ microstrip resonator (*blue*) constituting the bosonic oscillator mode. The readout resonator couples to the qubit capacitively and is read out with an open transmission line (TL) via the reflection signal of an applied microwave tone or pulse. The second resonator visible on chip is not used in the current experiment and is detuned in frequency from the relevant bosonic mode by ~0.5 GHz. The *scale bar* corresponds to 1 mm. **b** Effective circuit diagram of the device

The $\eta_1$-term is now the significant term and we move into its interaction picture. Satisfying the requirement $\omega_1 - \omega_2 = \eta_1$ and applying a RWA yields the effective Hamiltonian in the $\omega_1$ frame

$$\frac{\hat{H}_{\text{eff}}}{\hbar} = \frac{\eta_2}{2}\frac{\hat{\sigma}_z}{2} + \omega_{\text{eff}}\hat{b}^\dagger\hat{b} + \frac{g}{2}\hat{\sigma}_x\left(\hat{b}^\dagger + \hat{b}\right). \quad (4)$$

We define the effective bosonic mode energy $\omega_{\text{eff}} \equiv \omega - \omega_1$, which is the parameter governing the system dynamics. Noting $\eta_1 \gg \eta_2$, which is a necessary condition for the above approximation to hold, the effective qubit frequency $\eta_2$ and effective bosonic mode frequency $\omega_{\text{eff}}$ can be chosen as experimental parameters in the simulation. The complete coupling term of the quantum Rabi Hamiltonian is recovered, valid in the USC regime and beyond, while the geometric coupling strength is only modified by a factor of two, resulting in $g_{\text{eff}} = g/2$. It is therewith feasible to tune the system into a regime where the coupling strength is similar to or exceeds the subsystem energies. This is achieved by leaving the geometric coupling strength essentially unchanged in the synthesized Hamiltonian, while slowing down the system dynamics by effectively decreasing the mode frequencies to $\lesssim 8$ MHz. Thermal excitations of these effective transitions can be neglected since they couple to the thermal bath excitation frequency ~1 GHz of the cryostat via their laboratory frame equivalent frequency of $\omega_1/2\pi \sim 6$ GHz. We want to point out that the coupling regime is defined by $g_{\text{eff}}/\omega_{\text{eff}}$, rather than involving the Rabi frequency $\eta_1$, which does not enter the synthesized Hamiltonian. While the simulation scheme requires $|\epsilon - \omega_1| \ll \eta_1$, the qubit frequency does not enter the effective Hamiltonian. The time evolution of the qubit measured in the laboratory frame is subject to fast oscillations corresponding to the Rabi frequency $\eta_1$. Accordingly, the qubit dynamics in the engineered quantum Rabi Hamiltonian Eq. (4), valid in the $\omega_1$ frame, can be inferred from the envelope of the evolution in the laboratory frame. The derivation of Eq. (4) can be found in ref. [27] and is detailed in Supplementary Note 1. A similar drive scheme based on a Rabi tone was previously used in experiment to synthesize an effective Hamiltonian with a rotated qubit basis[31]. For the qubit and the bosonic mode degenerate in the laboratory frame, a distinct collapse-revival signature appears in the dynamics of the quantum Rabi model under USC conditions.

**Quantum simulation device.** The physical implementation of the quantum simulator is based on a superconducting circuit embedded in a typical circuit QED setup[32, 33], see Fig. 1. The atomic spin of the quantum Rabi model is mapped to a concentric transmon qubit[34, 35]. It is operated at a ratio of Josephson energy to charging energy $E_J/E_C = 50$ and an anharmonicity $\alpha/h = \omega_{12}/2\pi - \omega_{01}/2\pi = -0.36$ GHz $\sim -E_C/h = -0.31$ GHz, close to resonance with the bosonic mode at 5.948 GHz. $\omega_{ij}$ denote the transition frequencies between transmon levels $i$, $j$. The energy relaxation rate of the qubit at the operation point is measured to be $1/T_1 = 0.2 \times 10^6$ s$^{-1}$. An on-chip flux bias line allows for a fast tuning of the qubit transition frequency as the concentric transmon is formed by a gradiometric dc SQUID. The bosonic mode of the model is represented by a harmonic $\lambda/2$ resonator with an inverse lifetime $\kappa \sim 3.9 \times 10^6$ s$^{-1}$ that is limited by internal loss. Following the common convention, we use $\kappa$ as the inverse photon lifetime of a linear cavity, which may be extracted as the full width at half maximum of a resonance signature in frequency space. Via Fourier transformation one can see that this means the cavity relaxes to its groundstate at a rate of $\kappa/2$. In a separate experiment we find the internal quality of similar microstrip resonators to be limited to about $1.2 \times 10^4$ in the single photon regime, corresponding to a loss rate of $3.1 \times 10^6$ s$^{-1}$. Microwave simulations indicate that the quality is limited by radiation. The sample fabrication process is detailed in Supplementary Note 2.

**Sample characterization.** The quantum state collapse followed by a quantum revival is the most striking signature of the ultra-strong and close deep strong coupling regime of the quantum Rabi model and emerges for qubit and bosonic mode being degenerate in the laboratory frame. We calibrate this resonance condition by minimizing the periodic swap rate of a single excitation between qubit and bosonic mode for the simple Jaynes–Cummings model in the absence of additional Rabi drives. Figure 2 shows the measured vacuum Rabi fluctuations in the resonant case (a) and dependent on the qubit transition frequency (b). For initial state preparation of the qubit and readout we detune the qubit by 95 MHz to a higher frequency. This corresponds to switching off the resonant interaction with the bosonic mode. Supplementary Note 3 describes experimental details on flux pulse generation. Rabi vacuum oscillations can be observed during the interaction time $\Delta t$ and yield a coupling strength $g/2\pi = 4.3$ MHz, in good agreement with the spectroscopically obtained result, see Supplementary Note 8.

**Quantum state collapse and revival.** As the collapse-revival signature of the quantum Rabi model in USC conditions manifests most clearly for a vanishing qubit term, we initially set $\eta_2 = 0$, yielding the effective Hamiltonian in the qubit frame

$$\frac{\hat{H}}{\hbar} = \omega_{\text{eff}}\hat{b}^\dagger\hat{b} + \frac{g}{2}\hat{\sigma}_x\left(\hat{b}^\dagger + \hat{b}\right). \quad (5)$$

Figure 3a shows the applied measurement sequence which is based on the one in Fig. 2 but extended by a drive tone of amplitude $\eta_1$. The bosonic mode is initially in the vacuum state and the qubit is prepared in one of its basis states $|g\rangle$, $|e\rangle$, which are thermally impure. Qubit and bosonic mode are on resonance during the simulation time $\Delta t$. The drive is applied at a frequency $\omega_1$ detuned from the common resonance point by $\omega_{\text{eff}}$, setting the effective bosonic mode frequency in the rotating frame. Measured data for $\omega_{\text{eff}}/2\pi = 8$ MHz is displayed in Fig. 3b, corresponding to $g_{\text{eff}}/\omega_{\text{eff}} \sim 0.3$. Data points show the experimentally simulated time evolution of the qubit prepared in $|e\rangle$. A fast quantum state collapse followed by periodically returning quantum revivals can

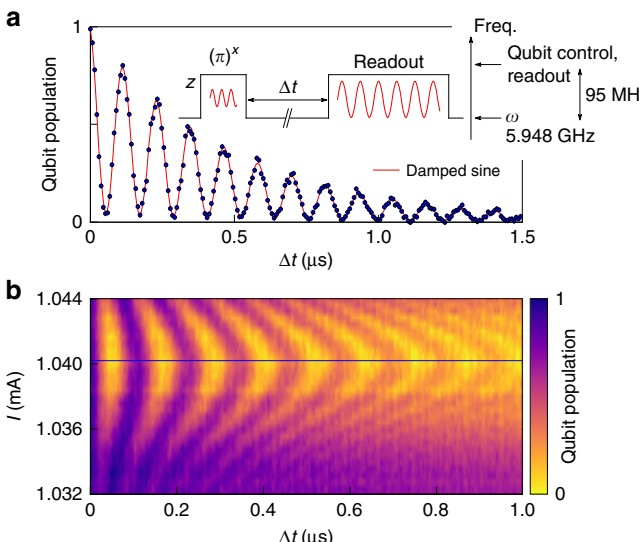

**Fig. 2** Vacuum Rabi oscillations between qubit and bosonic mode. **a** The qubit is initially dc-biased on resonance with the bosonic mode, while it is detuned for state preparation and readout. The *solid black line* in the inset depicts the fast flux pulses applied to the flux bias line and indicates the qubit frequency on the given axis. Qubit and bosonic mode are on resonance during an interaction time $\Delta t$. A frequency fit (*red*) of the vacuum Rabi oscillations yields $2g/2\pi = 8.5$ MHz. With the decay rate $\Gamma = (2.08 \pm 0.03) \times 10^6\ \text{s}^{-1}$ of the envelope and the qubit decay rate $1/T_1 = (0.2 \pm 0.12) \times 10^6\ \text{s}^{-1}$ we extract the bosonic mode decay rate $\kappa = (3.9 \pm 0.13) \times 10^6\ \text{s}^{-1}$. *Error bars* denote a statistical s.d. as detailed in the Methods. **b** For departing from the resonance condition (*blue line*) by varying the dc bias current $I$, we observe the expected decrease in excitation swap efficiency and an increase in the vacuum Rabi frequency. The qubit population is given in *colors* and we applied a numerical interpolation of data points

be observed. The ground state of the qubit subspace in the driven system as well as in the synthesized Hamiltonian, Eq. (5), is in the equatorial plane of the qubit Bloch sphere and is occupied after a time $\Delta t \gg T_1, 1/\kappa$. It is diagonal in the $|\pm\rangle$ basis, with $|\pm\rangle = 1/\sqrt{2}(|e\rangle \pm |g\rangle)$. The revival dynamics can be understood with an intuitive picture in the laboratory frame. The eigenenergies in the $|\pm\rangle$ subspaces take the form of displaced vacuum

$$\omega_{\text{eff}}\left(\hat{b}^\dagger \pm \frac{g}{2\omega_{\text{eff}}}\right)\left(\hat{b} \pm \frac{g}{2\omega_{\text{eff}}}\right) + \text{const.,} \qquad (6)$$

which is a coherent state that is not diagonal in the Fock basis. The prepared initial state in the experiment is therefore not an eigenstate in the effective basis with the drive applied such that many terms corresponding to the relevant Fock states $n$ of the bosonic mode participate in the dynamics with phase factors $\exp\{in\omega_{\text{eff}}t\}$, $n \in \mathbb{N}^+$. While contributing terms get out of phase during the state collapse, they rephase after an idling period of $2\pi/\omega_{\text{eff}}$ to form the quantum revival. The underlying physics of this phenomenon is fundamentally different from the origin of state revivals that were proposed for the Jaynes–Cummings model[36]. Here, the preparation of the bosonic mode in a large coherent state with $\alpha \gtrsim 10$ is required and non-periodic revivals are expected at times $\propto 1/g_{\text{eff}}$ rather than $\propto 1/\omega_{\text{eff}}$[37], as demonstrated in Supplementary Fig. 7. The blue line in Fig. 3b corresponds to a classical master equation simulation of the qubit dynamics in the rotating frame in the two-level approximation. It includes the second excited level of the transmon[38] and decay terms in the underlying Liouvillian according to measured values. Refer to Supplementary Note 5 for further details. Figure 3c, d

shows a classical simulation and the quantum simulation for $\omega_{\text{eff}}/2\pi = 5$ MHz with the qubit prepared in one of its eigenstates $|g\rangle$, $|e\rangle$. The population of the bosonic mode takes a maximum during the idling period and adopts its initial population at $2\pi/\omega_{\text{eff}}$ in the absence of dissipation, see Fig. 3e. The fast oscillations in Fig. 3c, d correspond to the Rabi frequency $\eta_1/2\pi \sim 50$ MHz. This value is chosen such that the requirement $\eta_1/\omega_{\text{eff}} \gg 1$ is fulfilled while staying well below the transmon anharmonicity, avoiding higher level populations. Deviations in the laboratory frame simulation traces are due to a uncertainty in the Rabi frequency that is extracted from Fourier transformation of measured data. The broadening in frequency space is mainly caused by the beating in experimental data, which is an experimental artifact. The relevant dynamics of the USC quantum Rabi Hamiltonian corresponds to the envelope of measured data. Since the laboratory frame dissipation is enhanced for a larger ratio of photon population in the bosonic mode, the accessible coupling regime is bound by the limited coherence of the bosonic mode, in particular. This is reflected in a dependence of the coherence envelope of the quantum revivals on the ratio $g/\omega_{\text{eff}}$, see Supplementary Fig. 7, reflecting that the excitation number is no longer a conserved quantity in the quantum Rabi model. We find a better agreement with experimental data for using a slightly increased value for the geometric coupling strength in the master equation simulation than extracted from vacuum Rabi oscillations. See Supplementary Notes 3 and 6 for a discussion and a summary of the relevant parameters.

The validity of the analog simulation scheme proposed in ref. [27] and used in this letter is confirmed by master equation simulations given in Supplementary Fig. 4. For ideal conditions, we demonstrate that the dynamics of the qubit and the bosonic mode in the quantum Rabi model is well reproduced by the constructed effective Hamiltonian and that the population of the bosonic mode is independent of the Rabi drive amplitude $\eta_1$, despite of it forming a large energy reservoir that is provided to the circuit.

In the experiment we face a parasitic coupling of the Rabi tones to the bosonic mode that is degenerate to the qubit and spatially close by in the circuit. This leads to an excess population of the bosonic mode, however without disturbing the functional evolution of its population. This is evident as the evolution of the simple harmonic Hamiltonian $\hat{H}_{\text{h}}/\hbar = \omega_{\text{eff}}\hat{b}^\dagger\hat{b} + \frac{1}{2}\eta_{\text{r}}(\hat{b}^\dagger + \hat{b})$ agrees with the expectation for the quantum Rabi model up to a scaling factor, where the last term corresponds to the parasitic drive of strength $\eta_{\text{r}}$ transformed to the rotating frame. By performing the displacement transformation $\hat{D} = \exp\{-\eta_{\text{r}}/(2\omega_{\text{eff}})(\hat{b}^\dagger - \hat{b})\}$, this contribution translates into a qubit tunneling term $\propto \hat{\sigma}_x$, giving rise to a sub-rotation of the effective frame. The resulting dynamics complies with the envelope defined by the ideal Hamiltonian with the tunneling term absent and therefore maps to the ideal quantum Rabi model, leaving its dynamics qualitatively unaffected. The transformations described are detailed in Supplementary Note 1, with master equation simulations supporting these statements in Supplementary Fig. 6. In Fig. 3d we made use of the topological symmetry of simulations with initial qubit states $|g\rangle$, $|e\rangle$, by subtracting two successive measurements with the qubit prepared in its eigenstates $|g\rangle$, $|e\rangle$, respectively, in order to cancel out the additional dispersive shift induced by the bosonic mode. As described in the Methods, we obtain the population evolution of the bosonic mode, depicted in Fig. 3e, by summing two successive measurements with the qubit prepared in $|g\rangle$, $|e\rangle$, respectively. We can infer its effective population by a fit to master equation simulations in the absence of a parasitic drive of the bosonic mode. Since the maximum population is around unity while the

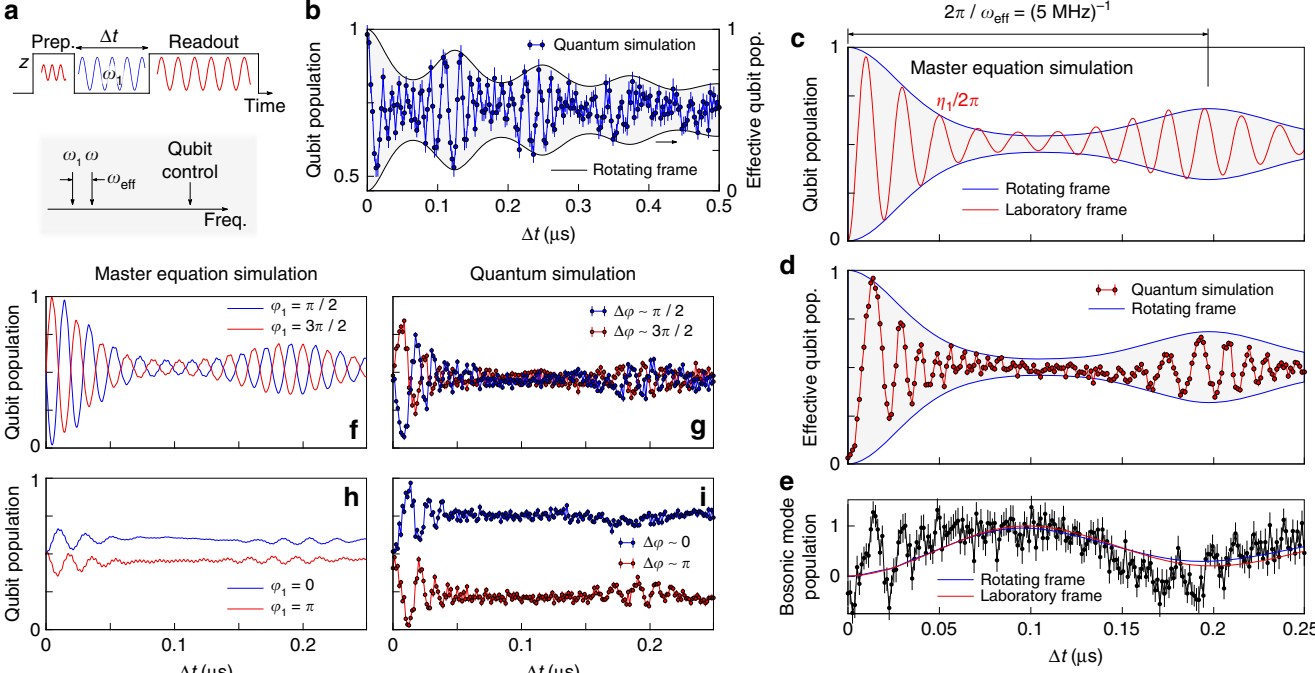

**Fig. 3** Quantum state collapse and revival with only the dominant Rabi drive applied. **a** Schematic pulse sequence and overview on the relative frequencies used in the experiment. **b** Quantum simulation of the periodic recurrence of quantum state revivals for $\omega_{eff}/2\pi = 8\,\text{MHz}$. The *blue line* corresponds to a master equation simulation of the qubit evolution in the rotating frame. **c, d** Master equation and quantum simulation of the qubit time evolution for initial qubit states $|g\rangle$, $|e\rangle$ and $\omega_{eff}/2\pi = 5\,\text{MHz}$, corresponding to $g_{eff}/\omega_{eff} \sim 0.5$. The *red line* shows the qubit population evolution of the driven system in the laboratory frame, Eq. (2), while the *blue lines* follow the qubit evolution in the synthesized Hamiltonian Eq. (4), likewise extracted from a classical master equation simulation. The deviation between the envelope of the laboratory frame data and the rotating frame data in **c** reflects the approximations of the simulation scheme. Experimental data shows the difference between two measurements for the qubit prepared in $|g\rangle$, $|e\rangle$, respectively, in order to isolate the qubit signal. **e** Measured population evolution of the bosonic mode, extracted from the sum of the two successive measurements and fitted to classically simulated data. **f–i** Qubit time evolution for varying relative phase $\varphi_1$ of the applied drive. The initial qubit state is prepared on the equator of the Bloch sphere $|g\rangle \pm |e\rangle$. Dispersive shifts induced by the bosonic mode are subtracted based on its classically simulated population evolution. *Error bars* throughout the figure denote a statistical s.d. as detailed in the Methods

qubit is in the equatorial state, the non-conservation of the total excitation number is apparent.

While the phase of the qubit Bloch vector is not well defined for initial states $|g\rangle$, $|e\rangle$, the qubit state carries phase information when prepared on the equatorial plane of the Bloch sphere via a $\pi/2$ pulse. Figure 3f–i shows the qubit time evolution with varying relative phase $\varphi_1$ between initial state and applied drive, plotted in the original qubit basis, as calibrated in a Rabi oscillation experiment. Experimentally, the orientation of the coordinate system is set by the first microwave pulse and we apply the Rabi drive with a varying relative phase $\varphi_1$, corresponding to the angle between qubit Bloch vector and rotation axis of the drive in the equatorial plane. When both are perpendicular, $\varphi_1 = \pm\pi/2$, similar oscillations including the state revival can be observed, assuming a steady state in the equatorial plane. For the case where $\varphi_1 = 0, \pi$, qubit oscillations in the laboratory frame are suppressed while the baseline is shifted up or down due to the detuning of the Rabi drive. The substructure emerges from the swap interaction term between qubit and bosonic mode that may be regarded as a perturbation as $\eta_1 \gg g$. Classical master equation simulations confirm that the basis shift, dependent on the prepared initial qubit state, is enhanced by the presence of the second excited transmon level and by a spectral broadening of the applied Rabi drive. The experimentally observed shift is not entirely captured by the classical simulation which we attribute to missing terms in the master equation that may be related to qubit tuning pulses and are unknown at present. See Supplementary Note 4 for a further discussion of the effect. Dependent on $\varphi_1$, we observe a

varying maximum photon population of the bosonic mode in classical simulations and indicated in the measured dispersive shift of the readout resonator. The qubit population as depicted in Fig. 3f–i is retrieved from measured raw data by subtracting the contribution of the bosonic mode. A deviation of the effective qubit basis is likewise observed for preparing the qubit in one of its eigenstates $|g\rangle$, $|e\rangle$.

**Full quantum Rabi model.** In order to simulate the full quantum Rabi model including a non-vanishing qubit energy term we switch on the second drive, $\eta_2 \neq 0$, see Fig. 4a. Quantum simulations are performed with the qubit initially in $|g\rangle$, subject to thermal excess population. The drive tones are up-converted in two separate IQ mixers while sharing a common local oscillator input to preserve their relative phase relation. For the simulation scheme to be valid, we need to fulfill the constraint $\omega_2 = \omega_1 - \eta_1$, see the schematics in Fig. 4b. This is achieved by initially applying a simulation sequence with $\eta_2 = 0$ in order to obtain the frequency equivalent of the Rabi frequency $\eta_1$ from a Fourier transformation of the qubit time evolution. Subsequently, we apply the same sequence with a finite $\eta_2$, $\varphi_1 = \varphi_2$ and $\omega_2$ set by obeying the above constraint. Figure 4c shows a master equation simulation of the complete quantum Rabi model for $\eta_2 = 0$ (*black*) and $\eta_2 \neq 0$ (*red*), respectively. The main difference is an emerging substructure between quantum revivals and an increase of the revival amplitude in the presence of the qubit energy term. The substructure before the first revival is not reproduced in

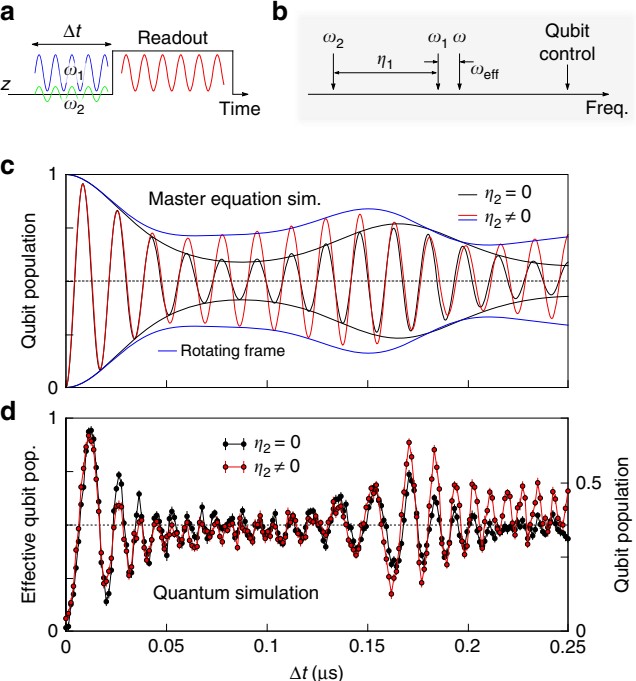

**Fig. 4** Simulation of the full quantum Rabi model. **a** Schematic pulse sequence used in the experiment. **b** Overview on the relative frequencies of the bosonic mode and the applied drives. The constraint $\eta_1 = \omega_1 - \omega_2$ is sketched. **c** Master equation simulations for vanishing qubit term $\eta_2 = 0$ (*black*) and with non-vanishing qubit term $\eta_2 > 0$ (*red*). The *blue line* corresponds to the classical simulation for $\eta_2/2\pi = 3$ MHz. **d** Quantum simulation for equal parameters. The dispersive shift of the readout resonator induced by the bosonic mode is subtracted based on classically simulated data. *Error bars* denote a statistical s.d. as detailed in the Methods

measured data, see Fig. 4d, which we attribute to ring up dynamics of the applied drives, such that the frequency constraint of the simulation scheme is not satisfied at small $\Delta t$. In addition, the parasitic drive of the bosonic mode contributes in parts to the suppression of the substructure. Convergence of the experimental simulation however can be observed better at later simulation times, where we observe an increase in the revival amplitude and more pronounced oscillations after the first revival, in agreement with the classical simulation. These signatures vanish in check measurements for intentionally violating the above constraint or applying the weak Rabi drive with a phase delay $\varphi_1 \neq \varphi_2$, see Supplementary Note 7. We estimate the frequency equivalent of $\eta_2/2\pi \sim 3$ MHz via comparing the relative peak heights of both drive tones with a spectrum analyzer. With $\omega_{eff}/2\pi = 6$ MHz we approach a regime where $2g_{eff}/\sqrt{\omega_{eff}\eta_2/2} > 1$, marking the quantum critical point in the related Dicke model[39].

The limitations imposed by the low coherence in the slowed down effective frame can be mitigated in a future experiment by employing a high-quality 3D cavity featuring a dc bias and a dedicated Rabi drive antenna coupling to the qubit. Fast tuning pulses may be realized by making use of the ac Stark shift induced by an off-resonant tone. A device with stronger suppression of parasitic couplings to the bosonic mode would not further require a classical post processing, which allows to extend the presented scheme to regimes where classical simulations become very inefficient.

## Discussion

We have demonstrated analog quantum simulation of the full quantum Rabi model in the ultra-strong and close deep strong coupling regime. The distinct quantum state collapse and revival signature in the qubit dynamics was observed, validating the experimental feasibility of the proposed scheme[27]. The main limitation of the scheme is an effective slowing down of the system dynamics, while the laboratory frame dissipation rates are maintained in the synthesized frame. In analogy to the measure of cooperativity in standard QED, we find the ratio $g_{eff}/\sqrt{\kappa/T_1} \sim 30$, rendering the qubit and bosonic mode decay rates an ultimate limitation for the simulation quality. The decelerated system dynamics in the effective frame however allows for the observation of quantum revivals on a timescale of ~100 ns, while the revival rate in the laboratory frame USC quantum Rabi model is on an sub-nanosecond scale, being experimentally hard to resolve. The small transmon anharmonicity limits the Rabi frequency to below ~100 MHz ~0.3 $|\alpha|$ in order to avoid higher level populations and suppress parasitic coupling to the bosonic mode. The accessible coupling regime is not limited by the simulation scheme, however we can experimentally observe quantum revivals only up to a coupling regime where $g_{eff}/\omega_{eff} \sim 0.6$ due to the finite coherence in our circuit.

While the presented dynamics can still be efficiently simulated on a classical computer, a true quantum supremacy will onset when incorporating more harmonic modes, leading to an exponential growth of the joint Hilbert space. Substituting the single-quantized mode by a continuous bosonic bath renders our setup a viable tool for investigating the spin boson model in various coupling regimes, which recently attracted experimental interest in the context of quantum simulations[40, 41]. The presented simulation scheme can be applied for a continuum of modes, such that an engineered bath in a restricted frequency band is collectively shifted by the applied Rabi drive frequency. This can become a route to address the infrared cutoff issue in a tailored bosonic bath and to observe a quantum phase transition in the spin boson model.

## Methods

**Experimental technique**. The quantum circuit is mounted in an aluminum box and cooled below ~50 mK. It is enclosed in a cryoperm case for additional magnetic shielding. Qubit preparation and manipulation microwave pulses are generated by heterodyne single sideband mixing and applied to the same transmission line used for readout. To ensure phase control of the drive tones with respect to the qubit Bloch sphere coordinate system fixed by the first excitation pulse, we use a single microwave source for qubit excitation and the drives required by the simulation scheme. Different pulses are generated by heterodyne IQ mixing with separate IQ frequencies and amplitudes. The bosonic mode resonator is located far away from the transmission line which reduces parasitic driving. Readout of the qubit state is performed dispersively by means of a separate readout resonator located at $\omega_r/2\pi$ = 8.86 GHz in a projective measurement of the $\hat{\sigma}_z$ operator with a strong readout pulse of 400 ns duration. Further details on the experimental setup are given in Supplementary Note 3.

**Protocol for extracting the qubit population**. In the simulation experiments presented in Figs. 3 and 4, we note a modulated low-frequency bulge in the recorded dispersive readout resonator shift that does not agree with the expected qubit population evolution. By comparing with the classical master equation simulation, we can recognize the population evolution of the bosonic mode which reflects the governing fundamental frequency $\omega_{eff}$ of the effective Hamiltonian. By simulating the full circuit Hamiltonian including qubit, bosonic mode and readout resonator, we find that the effect is induced by an additional photon exchange coupling $f$ between the bosonic mode and the readout resonator. The coupling is facilitated by the electric fields of the resonators and is potentially mediated by the qubit. See Supplementary Note 5 for the complete system Hamiltonian. In the diagonalized subspace of the two resonators, the bosonic mode can induce a cross-Kerr like photon number dependent shift $\propto f^2$ on the harmonic readout resonator as it inherits nonlinearity from the qubit. By adding or subtracting two subsequent simulation traces with the qubit prepared in either of the initial states $|g\rangle$, $|e\rangle$, we can isolate the signals corresponding to the population of the qubit and the bosonic mode. This measurement protocol is based on the symmetry of the qubit signal for preparing eigenstates, while the bosonic mode induced shift is always repulsive and does not change its sign. The photon exchange coupling $f$ therefore provides indirect access to the population of the bosonic mode without a dedicated readout device available. Specifically monitoring the population of the bosonic mode and

performing a Wigner tomography would highlight another hallmark signature of the USC regime, namely the efficient generation of non-classical cavity states[30]. Due to a lack of such a symmetry in case the relative phases of the Rabi drives are relevant, the qubit population can be retrieved from measured raw data based on the expectation for the bosonic mode population as obtained from the classical master equation simulation. In this procedure, the dispersive shift $\propto f^2$ remains as the only free fit parameter. See Supplementary Note 5 for more details on the described protocol.

**Data acquisition**. We readout the qubit state by observing the dispersive shift of the readout resonator which is acquired via a 400 ns long readout pulse. Full time traces, recording the readout pulse, are $2 \times 10^3$ fold pre-averaged per trace on our acquisition card. Successively, the data is sent to the measurement computer where we extract the IQ quadratures by Fourier transformation. We typically average over ~30 acquired traces to obtain a reasonable signal to noise ratio. Due to the reflection setup, most information is stored in the phase quadrature of the recorded signal. The given error bars represent the s.d. of the mean, as calculated from the pre-averaged data points and propagated according to Gauss.

**Data availability**. The data that support the findings of this study are available from the corresponding author upon reasonable request.

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

## Acknowledgements

We acknowledge valuable discussions with G. Romero, M.-J. Hwang, I. Pop, U. Vool, J. Pedernales and E. Solano. We are grateful to L. Radtke and S. Diewald for support during sample fabrication and A. Lukashenko for assistance in cryostat operation. This work was supported by the European Research Council (ERC) within consolidator Grant No. 648011 and through the KIT Nanostructure Service Laboratory (NSL). This work was also supported in part by the Ministry for Education and Science of the Russian Federation via NUST MISIS under contract K2-2016-063. J.B. acknowledges financial support by the Landesgraduiertenförderung (LGF) of the federal state Baden-Württemberg and by the Helmholtz International Research School for Teratronics (HIRST). A. Sch. acknowledges financial support by the Carl-Zeiss-Foundation.

## Author contributions

J.B. designed and fabricated the device, with input from M.W. and H.R. J.B. performed the measurements with support by A. Sch. and A.Ste. J.B. carried out data analysis and numerical simulations with contributions from M.M. J.B. wrote the manuscript with input from all coauthors. M.W. and A.U. supervised the project.

## Additional information

**Competing interests:** The authors declare no competing financial interests.

