## [Peer Review File · Nature Communications]

Reviewers' comments:

Reviewer #1 (Remarks to the Author):

Dear Editor,

this manuscript summarizes the design and experimental implementation of a quantum simulator of the Rabi model in the ultrastrong coupling regime. The Rabi model is the simplest Hamiltonian describing light-matter interaction, but it is usually found in nature in a regime where the coupling constant "g" is much smaller than the energies associated to the two-level system (matter) and light (the bosonic mode) themselves. This is for instance, the case of the starting point in this experiment with a transmon in a microwave resonator. But as the authors show, by driving transversely the qubit, it is possible to engineer a different effective dynamics, still of Rabi type, but in the USC regime.

The manuscript is very relevant to a broad spectrum of communities, from superconducting circuits, to quantum simulation, passing through quantum optics and photonics. The article is interesting and rather self-contained, with precise statements and clear indications that the USC regime has been achieved. I find that the work is an important milestone in the simulation of new light-matter interaction regimes, opening the door to studies with multilevel systems, quantum phase transitions and a variety of other applications.

I strongly recommend publication of this work in Nat. Comm. but I have a set of comments which the authors should address before publication. They relate to the presentation, explanation and interpretation of the measurements and are listed below without any particular order or priority.

* The first comment relates to the disappearance of the value " ϵ " in the effective dynamics. I understand that the simulation only requires $|\epsilon - \omega_1| \ll \eta_1$, but the authors should clarify this point (which is not clear in the original PRX, either).

* Reference 18 should be cited together with a similar experiment that also achieves USC coupling strengths, by P. Forn-Diaz and coworkers. It is true that this work, working with a zero-mode resonator, has less visually appealing RWA-violations, but the conclusions are as strong as in the Munich paper.

* The authors should discuss the role of thermal excitations in this setup. Are they negligible? The effective boson gap in the rotating frame is very small, ~ 5 MHz, and the qubit even has a close-to-zero frequency in 50% of the results. Are thermal excitations negligible in the lab frame, and thus need not be taken into account in the rotating frame? A justification of this ensues for a high-impact paper such as this.

* I must say that the experimental traces are a bit disappointing, in that they do not seem to be directly comparable to the simulations. At least, if I put Fig. 2c and 2d together they do not seem to be very similar inside the envelope: in particular, the experiment seems to involve more frequencies. This is not really discussed, and I had a hard time discerning whether the red lines in the experimental plots were coming from a classical simulation or were just a guide to the eye (which is what I now assume).

* The description of the measurement protocol at the beginning of page 4 is confusing and should have its own space in the Methods section. I would also suggest changing the letter λ and not call it a cross-Kerr effect: according to Eq. S19 from the supplementary material, this is a photon exchange term between resonators, which perhaps is mediated by the qubit (I do not know because

this is not elaborated). As such, it is not a cross-Kerr, whose meaning is phase interaction of the form $(a^\dagger a)(b^\dagger b)$ between number of photons.

* Continuing with this, in fig. S5 I do not understand the details. In "a", how does the experiment achieve different values of " λ "? In "c", what bosonic mode population is this? Readout or the simulation mode? In "d" how has the shift been extracted? Is this shift the red curve from fig. "b"?

* In the conclusions, the author explain that the upper limit for the available Rabi frequencies is 100 Mhz (at least for this transmon). I wonder why then the actual Rabi frequencies we observe are so much smaller, in the range of 6 MHz. Wouldn't larger values enhance the quality and robustness of the experiment?

Reviewer #2 (Remarks to the Author):

In their work, Braumuller et al. perform an analog quantum simulation of the Rabi model in a superconducting qubit chip. The scheme is implemented using two drive frequencies. In an appropriately rotating frame this makes the naturally available Jaynes-Cummings interactions behave as the full Rabi model.

The Rabi model is currently receiving much attention. Despite its long history, it has only recently been solved analytically. Although it has only two degrees of freedom, in another recent work it has been found to exhibit a quantum transition as a function of system parameters. And just a few months ago another experimental group presented a digital quantum simulation of the Rabi model using a different scheme (see Ref. [21]). For these reasons, the present work is very timely, and the topic is indeed interesting to a broader readership.

The paper presents quite a nice experiment, and an interesting step towards realizing Feynman's dream of quantum simulation in the platform of superconducting qubits. However I remain a bit reluctant to recommend publication of this article in Nature Communications, at least in the present form. First of all, there are quite some issues with the presentation, which I discuss further below and which have to be addressed. Secondly, I have some concerns regarding the role of the bosonic mode in the current experiment.

The Rabi model has but two degrees of freedom, one qubit and one harmonic oscillator. The present chip allows only for good access to one of the them. In fact, in the entire paper there is only a single measured curve for the boson population, which moreover seems quite noisy. State of the art experiments achieve much better measurements of the bosonic mode, even the measurement of its full Wigner function is possible.

Also, the Rabi model becomes interesting because the boson population can take very high values. In this work it is always at around 1 or below. In the Ref. [21] one can see measured photon numbers of up to 30. The reason is the limitation to $g_{\text{eff}}/\omega_{\text{eff}} < 0.7$, which is still not in the deep ultrastrong coupling. Maybe this says more about the the limitations of the scheme itself than about the quality of the experiment. Nevertheless for these small boson numbers one wonders how large the difference to the Jaynes-Cummings model really is. I think the authors can do a better job in conveying to a general readership the new physics contained in the Rabi model that they observe. Their main signature is the collapse and revival, but this is a phenomenon that appears also in the Jaynes Cummings model. The differences could be elaborated on much more clearly. Moreover the non-conservation of the total number of excitations could be much more emphasized. Here it is important to demonstrate that the deviation of total excitation number cannot be due to other

decoherence mechanisms, especially in view of the comment in lines 341-344 "In addition, a parasitic coupling of the applied drives to the bosonic mode gives rise to an excess population of the bosonic mode that eventually limits the simulation quality."

In addition the way the data is presented has some serious flaws and poses some questions. In my opinion these have to be addressed before a possible publication:

- There are no error bars on any of the data. These should be added.
- The qubit population should not be given in arbitrary units. Even if it is a difference between two experimental runs, it is still a number that can be calibrated.
- In several of the plots (for example Figure 2b), the data deviates very quickly from the theoretical predictions, but then the difference remains rather constant. Why is that so?
- The dependence of the observations on the model parameters could have deserved a more detailed analysis. Figures 3 and S6 are an effort in this direction, but there is very little in terms of quantitative comparison. What's more, the data for two values of η_2 in Figure 3 does not seem very convincing. Relevant figures of merit are the time scales of collapse and revival or the completeness of the collapse. For all of these the data seems to give a trend opposite to the theoretical prediction. I also find the explanation with the spectral broadening of the weak Rabi drive unsatisfactory. It may explain the fast deviation at short times, but I cannot see how the agreement to the ideal model could be bad at short times and improve later, except if the agreement is by chance, for example due to yet another undesired mechanism.
- The simulation happens in a rotating frame while the readout is performed in the laboratory frame. A phase-locked qubit rotation just before readout would allow one to observe the qubit state in a co-rotating frame. How difficult would it be to implement that experimentally?

Specific points which should help to improve the readability, listed in order as they appear in the text:

- Introduction: The state of the art of quantum simulation is referenced in a quite unsatisfactory way. The second paragraph gives the impression that there are only two experiments on analog quantum simulation outside of superconducting qubits. There exists a much larger body of works, and the choice of the two citations appears quite arbitrary. Trapped ions, where many quantum simulations have been performed and which are very relevant for the type of model simulated here (see my comment below on the Zitterbewegung), are not mentioned at all. To address a broader readership the article should be put into a broader context.
- It would be better to use different symbols for the Hamiltonians in Equations (1) and (2).
- After Equation (2), one reads "Within the RWA". Since there are several RWAs applied, it would be good to state which one.
- In arriving from Equation (2) to (3), it would be helpful to mention that terms rotating with $\exp(2i\omega_1 t)$ are neglected. In fact, without this, the statement "renders the first driving term time independent" is incorrect.
- Figure 1: The experiment in Figure 1 is for the Jaynes-Cummings model, Equation (2) with $\eta_1 = \eta_2 = 0$. While one may infer that from the description "Vacuum Rabi oscillation", stating that explicitly would help. Especially since just before the full Rabi model has been derived, so switching back to the Jaynes-Cummings model can be confusing for the reader.
- Figure 1: Why are there two different qubit readout points on the axis in the inset?
- Figure 1: The inset is not very well explained.
- Figure 1: $1/T_1$ and κ are not yet defined. Also Γ^{-1} is given by the inverse mean of $1/T_1$ and κ .
- Lines 169-170: "in agreement with the spectroscopically obtained result." It should be mentioned that this result is presented in the Supplementary Material.

- Lines 188-190: Since the fast oscillations are kind of confusing, it could be useful to emphasize again that one needs to look at the envelope.
 - Figure 2a: The x-axes have no labels.
 - Lines 221-223: "Dependent on the coupling regime, we experimentally create entangled states involving up to three photons." There is neither data that shows photon numbers significantly above 1 nor measurements of entanglement. How do the authors justify this statement?
 - Line 256-257: "which we attribute to missing higher order terms in the master equation" What is the nature of these terms?
 - Line 331: The parameter $2g_{\text{eff}}/\sqrt{\omega_{\text{eff}}\eta_2/2}$ is mentioned, but for a general reader its relevance in the context of the transition in the Rabi model should be mentioned.
 - Line 353-354: The Zitterbewegung has already been measured in a trapped-ion experiment: "Quantum simulation of the Dirac equation" by R. Gerritsma, G. Kirchmair, F. Zähringer, E. Solano, R. Blatt, C. F. Roos, Nature 463, 68 (2010). This paper should be cited. Also it shows that Hamiltonians that are very closely related to the Rabi model have been realized routinely in trapped ions, so that should also be mentioned.
 - Conclusions: The authors need classical input to be able to extract the desired signal from the measured data. For example in Figure 3 they subtract the calculated dispersive shift. In this light what are the prospects to actually go to regimes where classical calculations become impossible? Will such classical input present an obstacle in more complex situations?
- In the methods, it is not completely clear what times are meant for the decay, especially for the resonator mode. Is this the T1 time? The T2 time? The loss of bosons? Also, while for the qubit the authors give the coherence time, for the harmonic oscillator they give the decay rate. It would probably be best to stick to one of both, either times or rates.
- The different device and model parameters are scattered all over the paper and the supplementary material. It would be very helpful to have a table somewhere where all relevant parameters are listed together.

Supplementary Material

- Between Equations (S11) and (S12), the authors say "neglecting again time dependent terms". Since the \sin^2 terms are also time dependent, this should better read "time averaging".
- Figure S5: In the sentence "The difference between both simulations (gray) follows the simulated time evolution of the bosonic mode population (red), as obtained from the same simulation, and the dispersive shift extracted from experiment (blue)." the analog quantum simulation and numerical simulations should be distinguished.
- Figure S6: it is a bit confusing that the x-axis changes from plot to plot.

To conclude, the experiment is quite nice work addressing a timely topic and opening some interesting perspectives. Unfortunately however there is a long list of issues in the presentation and questions about the data, which together with my concerns about the role of the boson mode prevent me from recommending the article for Nature Communications, at least as it is.

Reviewer #3 (Remarks to the Author):

In this work the authors physically simulate the quantum Rabi model for very strong coupling by using a driving scheme to effectively 'slow down' the resonant frequencies of the system while leaving the 'spin'-oscillator coupling largely unchanged.

For works in this developing field of analog quantum simulation, I feel that the interest for a general

audience is two-fold, first in the actual results produced, and secondly in the potential for further development of more complicated systems. While the approach used in this work is interesting, one of the key features is a slowing of system dynamics relative to the coherence times of the system. I do not feel that the authors have adequately addressed this as an ultimate limitation of the system, or the true impact of decoherence in this work. Should the idea of 'cooperativity' from QED not apply here as a key measure, along side the strength of coupling?

This would allow the interested reader to compare this work to others and to future work. After all, even if the coupling is extremely strong, if the system loses coherence on the timescale of the expected revival periods (as is the case in the experiment here) it may rather strongly affect the dynamics compared to the pure coherent dynamics which are the stated goal of this protocol, and seems to be born out in the rather low contrast of the revival fringes in much of the data.

In the same vein, the authors have not given the reader enough information about the future potential, and how the rather short coherence times in this experiment may be remedied, especially in the proposed systems with more modes which will surely suffer from higher degrees of cross talk, etc. Mention is made of the anharmonicity of the transmon as a limitation, implying that a higher alpha is desirable for stronger driving, but this will come directly at the expense of decoherence due to charge noise for much larger alpha, which must be of concern.

Given these concerns, I believe this work is not of sufficiently broad applicability and interest for publication in this journal.

Reply to referee #1:

We appreciate your positive and very constructive feedback on our manuscript and addressed your comments by including the following changes:

Dear Editor,

this manuscript summarizes the design and experimental implementation of a quantum simulator of the Rabi model in the ultrastrong coupling regime. The Rabi model is the simplest Hamiltonian describing light-matter interaction, but it is usually found in nature in a regime where the coupling constant "g" is much smaller than the energies associated to the two-level system (matter) and light (the bosonic mode) themselves. This is for instance, the case of the starting point in this experiment with a transmon in a microwave resonator. But as the authors show, by driving transversely the qubit, it is possible to engineer a different effective dynamics, still of Rabi type, but in the USC regime.

The manuscript is very relevant to a broad spectrum of communities, from superconducting circuits, to quantum simulation, passing through quantum optics and photonics. The article is interesting and rather self-contained, with precise statements and clear indications that the USC regime has been achieved. I find that the work is an important milestone in the simulation of new light-matter interaction regimes, opening the door to studies with multilevel systems, quantum phase transitions and a variety of other applications.

I strongly recommend publication of this work in Nat. Comm. but I have a set of comments which the authors should address before publication. They relate to the presentation, explanation and interpretation of the measurements and are listed below without any particular order or priority.

** The first comment relates to the disappearance of the value " ϵ " in the effective dynamics. I understand that the simulation only requires $|\epsilon - \omega_1| \ll \eta_1$, but the authors should clarify this point (which is not clear in the original PRX, either).*

We clarified this point explicitly in the theory section and pointed out that the collapse-revival signature is only visible for qubit and bosonic mode degenerate in the laboratory frame.

** Reference 18 should be cited together with a similar experiment that also achieves USC coupling strengths, by P. Forn-Diaz and coworkers. It is true that this work, working with a zero-mode resonator, has less visually appealing RWA-violations, but the conclusions are as strong as in the Munich paper.*

We included the citation Forn-Diaz Sc. Reports 6 (2016) along with Niemczyk Nat. Phys. 6 (2010). We additionally pointed out that the specific spectral features of the USC regime were observed in these publications, while the onset of a breakdown of the RWA may have been encountered in other cQED experiments.

** The authors should discuss the role of thermal excitations in this setup. Are they negligible? The effective boson gap in the rotating frame is very small, $\sim 5\text{MHz}$, and the qubit even has a close-to-zero frequency in 50% of the results. Are thermal excitations negligible in the lab frame, and thus need not be taken into account in the rotating frame? A justification of this ensues for a high-impact paper such as this.*

Our experimental measurement temperature was typically in between 20mK – 50mK, corresponding to a thermal excitation frequency of roughly 0.4GHz – 1.0GHz. Experiments based on planar superconducting

circuits typically encounter groundstate qubit populations in excess of 5%, which is more than suggested by Planck's law and the excitation frequency stated above. A possible cause for this effect are quasiparticles that are created by infrared photons or cosmic particles. We pointed out in the manuscript that also in our case the qubit states are thermally impure, without stating an exact number. This leads essentially to an effective qubit basis with a smaller diameter of the Bloch sphere, reducing the signal to noise ratio when measuring the qubit population. This is the reason why we label our plots 'qubit population', where the distance between 0 and 1 corresponds to one complete population inversion we are able to achieve in the presence of our thermal background.

The effective transitions w_{eff} in the rotating frame that are in the few MHz regime or close to zero couple to the temperature bath of the cryostat in the laboratory frame via their lab frame equivalent frequency $w_{\text{eff}} + w_1$, where $w_1/2\pi \sim 6\text{GHz}$ is the transformation frequency to the rotating frame. The effective transitions in the rotating frame are therefore not thermalized within a timescale close to our simulation times or T_1 . This is different for the decay and decoherence, that indeed translates into the rotating frame. This becomes apparent in measured data due to the fact that while the effective timescale in the rotating frame is much slower, we still have to deal with decoherence at laboratory frequency timescales. We added a note on this subtlety in the manuscript.

** I must say that the experimental traces are a bit disappointing, in that they do not seem to be directly comparable to the simulations. At least, if I put Fig. 2c and 2d together they do not seem to be very similar inside the envelope: in particular, the experiment seems to involve more frequencies. This is not really discussed, and I had a hard time discerning whether the red lines in the experimental plots were coming from a classical simulation or were just a guide to the eye (which is what I now assume).*

The blue lines in Fig. 2c and 2d are identical and are the result of a master equation simulation of the ideal Hamiltonian to be synthesized. The authors added this remark in the figure caption. The red line in Fig. 2d is indeed a guide to the eye interconnecting data points, which is indicated by the symbol in the plot legend.

The Rabi frequency η_1 that corresponds to the fast oscillations visible in Fig. 2c is extracted from a Fourier transformation of the measured data, shown in red in Fig. 2d. Due to experimental noise and the small beating we encounter in the experiment, this frequency is not very sharp in frequency space, leading to the deviations you pointed out. The beating remains of unknown origin and is considered as an experimental artifact so far. We however want to point out that the relevant dynamics lies in the envelope of measured data, which is in reasonable agreement with the expectation, plotted in blue. We added all these remarks in the manuscript and also added a comment on the many frequency components of quantum simulated traces in the discussion.

** The description of the measurement protocol at the beginning of page 4 is confusing and should have its own space in the Methods section. I would also suggest changing the letter λ and not call it a cross-Kerr effect: according to Eq. S19 from the supplementary material, this is a photon exchange term between resonators, which perhaps is mediated by the qubit (I do not know because this is not elaborated). As such, it is not a cross-Kerr, whose meaning is phase interaction of the form $(a^\dagger b)$ between number of photons.*

We added a paragraph in the methods section, lining out the motivation and protocol of experimentally retrieving the evolution of the bosonic mode population. Accordingly, we rephrased the main points of the procedure in the main manuscript in order to remove ambiguities and avoid confusion.

We now call the coupling term between the resonators a photon exchange term throughout the manuscript, as you suggested, and changed its symbol to 'f'. We were motivated to call it a cross-Kerr coupling as it assumes the typical form $a^\dagger b^\dagger b$ after diagonalization in the resonator-resonator subspace. However, we agree that calling it cross-Kerr to begin with can be misleading. We noted this fact also

in the Supplemental Material and added a remark on the suspected coupling mechanism between the two resonators in the main manuscript.

** Continuing with this, in fig. S5 I do not understand the details. In "a", how does the experiment achieve different values of " λ "? In "c", what bosonic mode population is this? Readout or the simulation mode? In "d" how has the shift been extracted? Is this shift the red curve from fig. "b"?*

The authors updated axis labels and the caption of Fig. S5 in order to clarify its contents. Specifically, the plot in (a) shows classically simulated data and we achieve different values for the photon exchange coupling f by varying the coupling strength in the master equation simulation. The bosonic mode population in (c) is also classically simulated for realistic parameters and is the basis for subtracting the dispersive shift induced by the bosonic mode as performed in (d).

** In the conclusions, the author explain that the upper limit for the available Rabi frequencies is 100 Mhz (at least for this transmon). I wonder why then the actual Rabi frequencies we observe are so much smaller, in the range of 6 MHz. Wouldn't larger values enhance the quality and robustness of the experiment?*

It is true that a large value for the Rabi frequency enhances the quality of the simulation while there is an upper bound for the Rabi frequency set by the weak transmon anharmonicity and the mentioned parasitic coupling that increases with increasing drive strength. The Rabi frequencies used in the experiment are on the order of 50MHz, below the approximate threshold given in the conclusions in order to suppress effects affecting the simulation quality. The value is given in the manuscript (in line 228 of the original version). The frequency of around 6MHz is the effective bosonic mode frequency in the rotating frame, set by the detuning of the dominant Rabi drive with respect to the bosonic mode frequency in the laboratory frame.

Reply to referee #2:

We appreciate your very constructive feedback on our manuscript and addressed your remarks by including the following changes:

In their work, Braumuller et al. perform an analog quantum simulation of the Rabi model in a superconducting qubit chip. The scheme is implemented using two drive frequencies. In an appropriately rotating frame this makes the naturally available Jaynes-Cummings interactions behave as the full Rabi model.

The Rabi model is currently receiving much attention. Despite its long history, it has only recently been solved analytically. Although it has only two degrees of freedom, in another recent work it has been found to exhibit a quantum transition as a function of system parameters. And just a few months ago another experimental group presented a digital quantum simulation of the Rabi model using a different scheme (see Ref. [21]). For these reasons, the present work is very timely, and the topic is indeed interesting to a broader readership.

The paper presents quite a nice experiment, and an interesting step towards realizing Feynman's dream of quantum simulation in the platform of superconducting qubits. However I remain a bit reluctant to recommend publication of this article in Nature Communications, at least in the present form. First of all, there are quite some issues with the presentation, which I discuss further below and which have to be addressed. Secondly, I have some concerns regarding the role of the bosonic mode in the current experiment.

Thank you very much for pointing out this flaw in the data analysis. Motivated by your remark the authors performed a comprehensive analysis of the role of the bosonic mode in the experiment and thereby understood the impact of a parasitic driving of the bosonic mode induced by the drives required by the simulation scheme. We resolved this experimental side effect by demonstrating that the effective Hamiltonian remains in the form of the quantum Rabi model with an additional qubit tunneling term that does not alter the qualitative system dynamics. Please see our additional comments on this issue below that address your remarks specifically.

The Rabi model has but two degrees of freedom, one qubit and one harmonic oscillator. The present chip allows only for good access to one of the them. In fact, in the entire paper there is only a single measured curve for the boson population, which moreover seems quite noisy. State of the art experiments achieve much better measurements of the bosonic mode, even the measurement of its full Wigner function is possible.

The authors agree with your statement. Performing a Wigner tomography of the bosonic mode and demonstrating the presence of a non-classical cavity state would be another strong signature for the presence of ultra-strong coupling (USC), as was performed in Langford *et al.* This procedure however requires a sample with more complexity, containing an additional qubit used for mapping of the bosonic mode's density matrix and a rather sophisticated gate sequence for performing the Wigner tomography. We noted the benefit of an additional Wigner tomography for the experiment in the manuscript.

We argue that observing the collapse-revival signature in the qubit population evolution is a hallmark signature for the presence of USC, even without the support of an exact measurement of the bosonic mode population. While we currently do not have a dedicated readout device for the bosonic mode to our disposal, we encountered a photon exchange coupling between the (qubit) readout resonator and the bosonic mode which can be exploited to extract the form of the bosonic mode population and fit it to the expectation based on master equation simulations. Figure 2e has the purpose to demonstrate qualitative agree-

ment with the expected evolution, however we are not able to extract the photon number in the bosonic mode from the measurement, since the exact experimental value for the coupling strength between the two resonators is hard to calibrate. We noted this fact in the manuscript at describing Fig. 2e.

As a side remark, our research is aiming for investigating the spin boson model in a successive experiment, which reflects in the design and layout of the current sample. Since the single bosonic oscillator mode is to be replaced by a bath forming a continuous bosonic frequency band, a dedicated readout device for the bath would be much less straightforward.

Also, the Rabi model becomes interesting because the boson population can take very high values. In this work it is always at around 1 or below. In the Ref. [21] one can see measured photon numbers of up to 30. The reason is the limitation to $g_{\text{eff}}/\omega_{\text{eff}} < 0.7$, which is still not in the deep ultrastrong coupling. Maybe this says more about the the limitations of the scheme itself than about the quality of the experiment.

The limitation of the effective coupling ratio originates from the fact that the decoherence envelope becomes worse for an increasing photon number in the bosonic mode, which in turn increases for larger ratios $g_{\text{eff}}/\omega_{\text{eff}}$. As a consequence, the quality of the bosonic mode is one important limiting factor for the accessible coupling regime and the simulation quality. We pointed out this important fact in the manuscript and elaborated on this limitation in our conclusions. In the end, we are limited by the internal quality of the microstrip resonator that forms the bosonic mode, where radiation is one of the main loss contributions. We added this remark in the manuscript when discussing the decay rate of the bosonic mode and picked it up in the conclusions.

Concerning the classification of the coupling regime, the authors followed the original figures of merit, given for example in Casanova *et al.* PRL 105 (2010). USC onsets when the coupling becomes large enough such that the RWA clearly breaks down and the threshold is typically set at $g/w \sim 0.1$ on resonance. The deep strong coupling regime starts at a value $g = w$ and even a perturbative treatment of the RWA fails. Based on these definitions, the effective Hamiltonians in our experiments with $g/w \sim 0.5$ and up to $g/w \sim 0.6$ are well within the regime of USC.

Having said that, the interesting coupling regime of the quantum Rabi model is the USC regime as well as the deep strong coupling regime for $g/w \lesssim 10$, where the discussed non-trivial physics emerges. For even higher ratios g/w , the energy terms in the Hamiltonian can be efficiently treated perturbatively and the Hamiltonian eventually takes a trivial form again after a Hadamard rotation. Highest photon numbers appear at $g/w \rightarrow \infty$, which is the trivial case where a resonant Rabi drive indirectly populates the bosonic mode up to a photon decay equilibrium population. The population number alone is therefore not a good figure of merit for the non-classicality of the observed physics.

Nevertheless for these small boson numbers one wonders how large the difference to the Jaynes-Cummings model really is. I think the authors can do a better job in conveying to a general readership the new physics contained in the Rabi model that they observe. Their main signature is the collapse and revival, but this is a phenomenon that appears also in the Jaynes Cummings model. The differences could be elaborated on much more clearly.

While we cannot directly access the population of the bosonic mode in the experiment, we rely on the collapse-revival signature in the qubit population in order to claim the presence of USC in our effective Hamiltonian. Assigning the observed dynamics to USC physics is justified by the following reasons: The initial state we prepare is the vacuum in the bosonic mode and the qubit in its ground state. This is a ground state of the Jaynes Cummings model, but not a ground state of the quantum Rabi model in the USC regime, compare Casanova, ..., Solano PRL 105 (2010). During the simulation, the qubit state evolves to its new ground state while showing the characteristic collapse-revival dynamics, which can be understood with the

intuitive picture presented in the manuscript. To the best of our knowledge, a similar collapse revival signature in the qubit population for the Jaynes Cummings model (with a RWA applied) was proposed only for preparing a large coherent state in the cavity mode ($\alpha > 10$), see Eberly, Narozhny, and Sanchez-Mondragon PRL 44 (1980). In addition, the revivals appears non-periodic and at times proportional to the Vacuum Rabi frequency g_{eff} , rather than the effective bosonic mode frequency w_{eff} . The positions of the revivals observed in our experiment are demonstrated to agree with the theoretical expectation for the USC regime in Fig. 2,3 and S7.

The authors added these remarks and discussions in the manuscript in order to convey the important distinction to the reader.

Moreover the non-conservation of the total number of excitations could be much more emphasized. Here it is important to demonstrate that the deviation of total excitation number cannot be due to other decoherence mechanisms, especially in view of the comment in lines 341-344 "In addition, a parasitic coupling of the applied drives to the bosonic mode gives rise to an excess population of the bosonic mode that eventually limits the simulation quality."

Thank you for pointing out that this information was not emphasized enough when discussing the data. Since we do not access the bosonic mode population in the experiment directly, we demonstrate the non-conservation of the total excitation number based on our master equation simulations both in the laboratory frame and the rotating frame for varying ω_{eff} .

Motivated by your concerns on the role of the bosonic mode, we reconsidered the influence of the parasitic driving that can excess populate the bosonic mode in the experiment. While we cannot easily calibrate the exact strength η_r of this parasitic coupling, we find that it translates into a σ_x term in the otherwise ideal effective Hamiltonian. The effective excitation number in the bosonic mode therefore follows the theoretical expectation and we therefore quantitatively state the numbers from our master equation simulations while demonstrating a functional agreement of the population evolution with the expectation.

We added this analysis and discussion in the manuscript and detailed the underlying calculations in the Supplemental Material. In addition, we added classical simulations including a realistic σ_x term in the rotating frame Hamiltonian, demonstrating a small quantitative deviation.

The simulations shown in Fig. 2e were performed with a small realistic parasitic coupling switched on, in order to reproduce experimental conditions, see Sec. V in the Supplemental Material. However, we checked that the population is not altered significantly for the parasitic coupling switched off, which means that the given value on the axis of Fig. 2e corresponds to the true population of the bosonic mode and is not enhanced by other mechanisms. The simulations in the rotating frame in Fig. S6 also give the true theoretical bosonic mode population.

In addition the way the data is presented has some serious flaws and poses some questions. In my opinion these have to be addressed before a possible publication:

- There are no error bars on any of the data. These should be added.

The authors added error bars on all of the experimental data. Errors represent the standard deviation of the mean, accounting for statistical errors of pre-averaged data. We added a paragraph in the Methods that describes the origin of data points and the meaning of the error bars.

- The qubit population should not be given in arbitrary units. Even if it is a difference between two experimental runs, it is still a number that can be calibrated.

We updated axis labels such that the qubit population is given in the calibrated Rabi frame or in the effective qubit frame. The change in effective qubit frame is discussed in the manuscript and the Supplemental Material.

- In several of the plots (for example Figure 2b), the data deviates very quickly from the theoretical predictions, but then the difference remains rather constant. Why is that so?

For small simulation times, we have to deal with ring-up dynamics in our drive pulses and the frequency of very short pulses is not a well-defined quantity, which disturbs the simulation quality at small simulation times. We now consider this as the main reason for the poor simulation quality in Fig. 3, at simulation times smaller than 150ns. The authors added this discussion in the manuscript.

- The dependence of the observations on the model parameters could have deserved a more detailed analysis. Figures 3 and S6 are an effort in this direction, but there is very little in terms of quantitative comparison. What's more, the data for two values of η_2 in Figure 3 does not seem very convincing. Relevant figures of merit are the time scales of collapse and revival or the completeness of the collapse. For all of these the data seems to give a trend opposite to the theoretical prediction. I also find the explanation with the spectral broadening of the weak Rabi drive unsatisfactory. It may explain the fast deviation at short times, but I cannot see how the agreement to the ideal model could be bad at short times and improve later, except if the agreement is by chance, for example due to yet another undesired mechanism.

We added another figure in the Supplemental Material to address this point and discuss the influence of disruptive experimental effects in more detail in the manuscript. The authors agree that the quality of the experimental simulation is clearly limited for the measurements in Fig. 3 and point out this fact in the manuscript. We added a more extensive discussion of the observed signatures and provided realistic reasoning for the deviations from the master equation simulation. We removed the argument with the spectral broadening as you suggest and mention the issue of ring-up dynamics and an uncertainty in the underlying Fourier transformation. We argue that the frequencies of the short drive pulses at early times are not well defined and therefore the constraint required by the simulation scheme is not well satisfied. This condition is met much better for later times, where the expected signatures appear. To cross check that this is not by chance and to gain more confidence about these results, we performed measurements with intentionally violated parameter constraints and demonstrate that the expected signatures are suppressed. In parts, the suppression of the substructure between revivals is caused by the discussed parasitic drive of the bosonic mode, according to master equation simulations.

- The simulation happens in a rotating frame while the readout is performed in the laboratory frame. A phase-locked qubit rotation just before readout would allow one to observe the qubit state in a co-rotating frame. How difficult would it be to implement that experimentally?

Given the small beating we encounter in the experimental curves and the mentioned ring-up dynamics in the applied drives, keeping track of the phase of the qubit seems to be rather challenging. Especially for simulations including the second Rabi drive, the authors are hesitant to believe that the suggested qubit rotation is feasible in the current experiment under continuous driving. However, in particular for small Rabi frequencies, the information in the envelope can substantially increase and we will investigate the viability of the proposed protocol in future experiments.

Specific points which should help to improve the readability, listed in order as they appear in the text:

- Introduction: The state of the art of quantum simulation is referenced in a quite unsatisfactory way. The second paragraph gives the impression that there are only two experiments on analog quantum

simulation outside of superconducting qubits. There exists a much larger body of works, and the choice of the two citations appears quite arbitrary. Trapped ions, where many quantum simulations have been performed and which are very relevant for the type of model simulated here (see my comment below on the Zitterbewegung), are not mentioned at all. To address a broader readership the article should be put into a broader context.

Thank you for noting that we did an unsatisfactory job in presenting a comprehensive list of publications in the field of analog quantum simulations. We updated the paragraph in order to make the collection of mentioned works more exhaustive and in particular mention the contributions from the field of trapped ions.

- It would be better to use different symbols for the Hamiltonians in Equations (1) and (2).

We changed the symbol of the Hamiltonian in Eq. (2) to \hat{H}_d .

- After Equation (2), one reads "Within the RWA". Since there are several RWAs applied, it would be good to state which one.

We added the remark that we consider a rotating wave approximation that holds when $\eta_i/\omega_i \ll 1$.

- In arriving from Equation (2) to (3), it would be helpful to mention that terms rotating with $\exp(2i\omega_1 t)$ are neglected. In fact, without this, the statement "renders the first driving term time independent" is incorrect.

The authors corrected for this impreciseness and added the remark in the manuscript as you suggested.

- Figure 1: The experiment in Figure 1 is for the Jaynes-Cummings model, Equation (2) with $\eta_1 = \eta_2 = 0$. While one may infer that from the description "Vacuum Rabi oscillation", stating that explicitly would help. Especially since just before the full Rabi model has been derived, so switching back to the Jaynes-Cummings model can be confusing for the reader.

We added the suggested remark in order to remove any ambiguities.

- Figure 1: Why are there two different qubit readout points on the axis in the inset?

The label 'readout' was indicating the location of the readout resonator. The authors removed this label now in order to avoid confusion and since this information is not relevant for the figure.

- Figure 1: The inset is not very well explained.

We added some additional explanations in order to make the inset clearer.

- Figure 1: $1/T_1$ and κ are not yet defined. Also Γ^{-1} is given by the inverse mean of $1/T_1$ and κ .

We now define all quantities and corrected the mistake you pointed out by restricting to only quote and compare decay rates.

- Lines 169-170: *“in agreement with the spectroscopically obtained result.” It should be mentioned that this result is presented in the Supplementary Material.*

We added the reference to the Supplemental Material as you suggested.

- Lines 188-190: *Since the fast oscillations are kind of confusing, it could be useful to emphasize again that one needs to look at the envelope.*

We emphasized this fact in the same paragraph a bit further down, as it applies for Fig. 2d as well.

- Figure 2a: *The x-axes have no labels.*

We updated the figure and added time and frequency labels, respectively.

- Lines 221-223: *“Dependent on the coupling regime, we experimentally create entangled states involving up to three photons.” There is neither data that shows photon numbers significantly above 1 nor measurements of entanglement. How do the authors justify this statement?*

In light of the reworked role of the bosonic mode, we removed this remark, in agreement with our argumentation on the bosonic mode population based on the master equation simulations.

- Line 256-257: *“which we attribute to missing higher order terms in the master equation” What is the nature of these terms?*

Thank you for pointing out that this formulation was misleading. The authors corrected the statement in the manuscript and point out that the effect is attributed to missing terms in the master equation with nature and origin unknown at present. We conjecture that the effect as an experimental artifact that may be related to the fast frequency tuning pulses, as indicated in the main text and the Supplemental Material.

- Line 331: *The parameter $2g_{\text{eff}}/\sqrt{\omega_{\text{eff}}\eta_2/2}$ is mentioned, but for a general reader its relevance in the context of the transition in the Rabi model should be mentioned.*

We included a citation to Nataf and Ciuti Nat. Comm. 1 (2010) together with the remark that the condition marks the quantum critical point in the related Dicke model.

- Line 353-354: *The Zitterbewegung has already been measured in a trapped-ion experiment: “Quantum simulation of the Dirac equation” by R. Gerritsma, G. Kirchmair, F. Zähringer, E. Solano, R. Blatt, C. F. Roos, Nature 463, 68 (2010). This paper should be cited. Also it shows that Hamiltonians that are very closely related to the Rabi model have been realized routinely in trapped ions, so that should also be mentioned.*

Thank you for pointing out that we missed citing this publication which is highly related and relevant for the experiment presented. We mentioned the paper in the introduction and again explicitly when discussing the Zitterbewegung in the conclusions.

- Conclusions: *The authors need classical input to be able to extract the desired signal from the measured data. For example in Figure 3 they subtract the calculated dispersive shift. In this light what are the prospects to actually go to regimes where classical calculations become impossible? Will such classical input present an obstacle in more complex situations?*

In the current experiment, which we consider a proof-of-principle experiment, demonstrating the applicability of Hamiltonian engineering for analog quantum simulation, we performed a classical post processing in order to compensate for experimental limitations. In an improved version of the device, with decreased parasitic coupling between bosonic mode and the drive tones as well as the readout resonator, classical input becomes unnecessary. In an improved future device, we would employ a high quality 3D cavity mode, and a qubit that can be dc tuned via an external coil. Fast frequency pulsing of the qubit may be achieved by making use of the ac Stark shift. The authors discuss how to mitigate major experimental limitations now in the conclusions of the manuscript and added a note that classical post processing would not be further required.

- In the methods, it is not completely clear what times are meant for the decay, especially for the resonator mode. Is this the T_1 time? The T_2 time? The loss of bosons? Also, while for the qubit the authors give the coherence time, for the harmonic oscillator they give the decay rate. It would probably be best to stick to one of both, either times or rates.

The authors state all quantities as real rates with unit 1/s now and removed ambiguities by adding the term 'energy relaxation.' We clarified in the manuscript that we use κ as the inverse photon lifetime of the cavity.

- The different device and model parameters are scattered all over the paper and the supplementary material. It would be very helpful to have a table somewhere where all relevant parameters are listed together.

Thank you very much for the suggestion. We added two tables in the Supplemental Material summarizing the relevant device and simulation parameters.

Supplementary Material

- Between Equations (S11) and (S12), the authors say "neglecting again time dependent terms". Since the \sin^2 terms are also time dependent, this should better read "time averaging".

We corrected for this impreciseness as you suggest.

- Figure S5: In the sentence "The difference between both simulations (gray) follows the simulated time evolution of the bosonic mode population (red), as obtained from the same simulation, and the dispersive shift extracted from experiment (blue)." the analog quantum simulation and numerical simulations should be distinguished.

We updated the figure and its caption in order to remove the ambiguity you mentioned.

- Figure S6: it is a bit confusing that the x-axis changes from plot to plot.

We moved part (f) to the previous figure and omitted part (e) due to repetition. We have chosen measurement windows in the experiment such that the relevant information is captured while saving acquisition time. For panel (d), where the first revival appears only after 250ns, the window was chosen larger than for simulations with higher ω_{eff} . We added a note in the caption to avoid confusion for the reader.

To conclude, the experiment is quite nice work addressing a timely topic and opening some interesting perspectives. Unfortunately however there is a long list of issues in the presentation and questions

about the data, which together with my concerns about the role of the boson mode prevent me from recommending the article for Nature Communications, at least as it is.

Reply to referee #3:

Thank you for your feedback on our manuscript. We tried to address your concerns and remarks by including the following changes in our manuscript:

In this work the authors physically simulate the quantum Rabi model for very strong coupling by using a driving scheme to effectively 'slow down' the resonant frequencies of the system while leaving the 'spin'-oscillator coupling largely unchanged.

For works in this developing field of analog quantum simulation, I feel that the interest for a general audience is two-fold, first in the actual results produced, and secondly in the potential for further development of more complicated systems. While the approach used in this work is interesting, one of the key features is a slowing of system dynamics relative to the coherence times of the system. I do not feel that the authors have adequately addressed this as an ultimate limitation of the system, or the true impact of decoherence in this work. Should the idea of 'cooperativity' from QED not apply here as a key measure, along side the strength of coupling?

The authors realized that the point of effectively slowing down the system dynamics was not made clear enough. We address it now already in the theory part when introducing the simulation scheme and describing how the USC condition is synthesized in the effective frame. In addition, we renewed the first paragraph in the discussion, pointing out that the finite lifetime of the system is the main limiting factor for the simulation quality. At the same time, we point out that the decelerated system dynamics leads to a revival rate that is experimentally well accessible, while a laboratory frame USC quantum Rabi system would show repeated revivals on a sub-nanosecond scale.

We added the analogy to the cooperativity measure from QED and find $C=g/\sqrt{\kappa/T_1}\sim 30$, quantifying the influence of loss for our simulation and showing that the decay rates are ultimately limiting the quality of the experiment.

This would allow the interested reader to compare this work to others and to future work. After all, even if the coupling is extremely strong, if the system loses coherence on the timescale of the expected revival periods (as is the case in the experiment here) it may rather strongly affect the dynamics compared to the pure coherent dynamics which are the stated goal of this protocol, and seems to be born out in the rather low contrast of the revival fringes in much of the data.

The authors agree with your statement that decoherence does affect the system at a timescale where the revival is expected. Nevertheless, the collapse-revival signature is well pronounced and agrees well with the dynamics as expected from master equation simulations. We are confident that the observed dynamics originates from USC physics as the positions of the revivals follow the expectation according to $1/\omega_{\text{eff}}$.

We regard the current experiment as a proof-of-principle, demonstrating the experimental feasibility of the simulation scheme in order to investigate USC physics by qualitatively showing reasonable agreement of the experimental findings with master equation simulations. These signatures are of unique quantum origin and to the best of our knowledge were not experimentally observed before except for in the related work by Langford *et al.* Beyond the physics content of our experiment, as you pointed out in your comments, the reader may find it interesting to see one of the first experiments on analog quantum simulation with superconducting qubits.

In a future experiment, one may employ a 3D cavity with a quality factor exceeding 10^6 , and a qubit that can be dc tuned via an external coil. Fast frequency pulsing of the qubit may be achieved by making use of

the ac Stark shift. The authors have included a small outlook in the manuscript, discussing how to mitigate major experimental limitations.

In the same vein, the authors have not given the reader enough information about the future potential, and how the rather short coherence times in this experiment may be remedied, especially in the proposed systems with more modes which will surely suffer from higher degrees of cross talk, etc.

The authors reworked the conclusions, addressing the future potential of the presented simulation scheme in more detail and discussing the influence of the rather small cooperativity ratio. As stated above, we also offer a route to mitigate this issue in a future experiment.

The proposed spin boson model is a very interesting system to investigate with quantum simulators since it involves many degrees of freedom in a vast Hilbert space, rendering the system hard to trace classically even with few harmonic modes (in the limit of strong coupling). We point out in the manuscript that the simulation scheme presented for the quantum Rabi model can be applied to the more complex spin boson model as well, which is not obvious for digital schemes that may require phase corrected pulses for individual modes.

In the present manuscript it was our goal to simulate the quantum Rabi model in the USC regime. In this case the overall decay envelope in the simulator is defined by the decay rates of the qubit and the bosonic mode. However, the situation is very different in case of the spin boson model. Here, we are interested in the dissipative dynamics of the qubit in the presence of an intentionally dissipative bosonic bath with certain spectral function. Only the finite coherence of the qubit is a limiting feature, while decay and decoherence from the bosonic modes is considered as part of the simulation.

We added a remark on this issue in the last paragraph of the conclusions.

Mention is made of the anharmonicity of the transmon as a limitation, implying that a higher alpha is desirable for stronger driving, but this will come directly at the expense of decoherence due to charge noise for much larger alpha, which must be of concern.

It is correct that the simulation quality would in principle benefit from a higher drive amplitude. The transmon will start to depart drastically from its well-known insensitivity to charge fluctuations at an E_J/E_C ratio of ~ 20 , leaving us a little margin for further increasing its anharmonicity. In the current experiment, an upper limit to the Rabi frequency is also set by the parasitic coupling of the drive to the bosonic mode. This issue can be addressed in a 3D cavity approach by careful field engineering and antenna layout. Depending on the coupling regime we aim for, the requirement $\eta_1/\omega_{\text{eff}} \gg 1$ is fulfilled in the current experiment and can be maintained in future similar experiments. We added a remark on both upper and lower thresholds for η_1 in the manuscript.

Given these concerns, I believe this work is not of sufficiently broad applicability and interest for publication in this journal.

REVIEWERS' COMMENTS:

Reviewer #1 (Remarks to the Author):

Dear Editor, I have read the revised manuscript as well as the reply letter. As anticipated in my original report, I think this work is a very nice contribution to the fields of quantum simulation and quantum optics, opening a new avenue for the implementation of strongly correlated models with circuits. The authors have addressed all my minor comments and the new manuscript should be published in its present form.

Reviewer #2 (Remarks to the Author):

I very much appreciate the detailed and serious response to all the three Referees' comments. The paper has been thoroughly refurbished, and, in as far as the limitations of the device allow, the response to the raised concerns seem satisfactory.

Given that the paper is now much better accessible for a general readership and that a variety of issues have been cleared (e.g., the role of the boson mode, experimental errors and units, clarification of relevant experimental signatures), I am now much more in favor of publication. While the device is clearly limited in its performance, the experiment does show the potential of superconducting qubits for quantum simulation, at least as a proof of principle, and as such can be a good fit for Nature Communications.

Reviewer #4 (Remarks to the Author):

In the manuscript "Analog quantum simulation of the Rabi model in the ultra-strong coupling regime" Braumüller et al. use a transmon qubit coupled to a superconducting resonator to implement an analog quantum simulation that mimics the quantum Rabi model. In order to do this, the authors apply a 2-tone drive that creates an effective Hamiltonian in which the dynamics of the qubit and the mode are slowed down, while their coupling and coherence times remain almost unchanged. With appropriately chosen drive frequencies and amplitudes this effective Hamiltonian reproduces the dynamics of the Rabi model in the ultra-strong coupling regime, and the authors observe a signature of this regime, the collapse and revival of the quantum state, in experiment.

The manuscript is well written, the data is convincing and the main text and supplementary information contain an extensive characterization of the conducted experiments. The topic of analog quantum simulations is timely and of interest to a broad audience, and the manuscript should definitely be published. This being said, I do have some concerns regarding the novelty of the manuscript and whether it represents an important advance of significance to the field:

- The ultra-strong coupling regime has already been reached directly in Refs. [12] & [13]. Are there any particularly interesting locations in parameter space that can be much easier reached in the analog simulations than in the direct experiment, i.e. how useful is the simulation of this particular problem?
- Although the manuscript by Braumüller et al. goes through the derivation of the effective quantum Rabi Hamiltonian in the main text and even more detail in the supplementary material, the idea and theory are entirely based on Ref. [25].
- The experimental implementation of the two-tone drive and the creation of the effective Hamiltonian has already been demonstrated in Ref. [26], although for different parameters and with a different motivation.

In the light of the extensive previous comments from 3 referees in the first rounds of reviews, I will refrain from making more technical comments about the manuscript. However, I have been asked to comment on the reply to referee #3:

- When reading through the manuscript, I have found that the slowing down of the system dynamics relative to the coherence times of the system has been clearly introduced in this version.
- Referee #3 has made the valid point that the revival signature occurs at the same timescale as the system's coherence times. This makes it more difficult to observe the pure coherent dynamics, however as the authors point out, the collapse-revival signature is strong enough to originate convincingly from the USC physics.
- The authors have also added sufficient information about the future potential and how longer coherence times can be achieved in the experiment.
- Also the comment about the anharmonicity of the transmon being a limitation has been satisfactorily addressed.

Overall, I consider this manuscript a very strong and interesting piece of work. Although it might not contain any new or unexpected physics, it is still a proof-of-principle demonstration of the experimental feasibility of this simulation scheme. Hence, it will definitely be of high interest to specialists in the fields of superconducting cavity quantum electrodynamics and analog quantum simulation.

Reply to referee #4:

We appreciate your positive feedback on our manuscript and addressed your comments as follows:

In the manuscript "Analog quantum simulation of the Rabi model in the ultra-strong coupling regime" Braumüller et al. use a transmon qubit coupled to a superconducting resonator to implement an analog quantum simulation that mimics the quantum Rabi model. In order to do this, the authors apply a 2-tone drive that creates an effective Hamiltonian in which the dynamics of the qubit and the mode are slowed down, while their coupling and coherence times remain almost unchanged. With appropriately chosen drive frequencies and amplitudes this effective Hamiltonian reproduces the dynamics of the Rabi model in the ultra-strong coupling regime, and the authors observe a signature of this regime, the collapse and revival of the quantum state, in experiment.

The manuscript is well written, the data is convincing and the main text and supplementary information contain an extensive characterization of the conducted experiments. The topic of analog quantum simulations is timely and of interest to a broad audience, and the manuscript should definitely be published. This being said, I do have some concerns regarding the novelty of the manuscript and whether it represents an important advance of significance to the field:

- The ultra-strong coupling regime has already been reached directly in Refs. [12] & [13]. Are there any particularly interesting locations in parameter space that can be much easier reached in the analog simulations than in the direct experiment, i.e. how useful is the simulation of this particular problem?

The USC regime was reached in Refs. [12] and [13] directly by sample design. By investigating possible parameter regimes for geometric USC, one can find certain natural limitations. For instance the transmon qubit will break out of the phase regime when approaching USC while maintaining its transition frequency, and essentially become a Cooper pair box. This general no-go theorem to achieve USC with a transmon circuit was discussed for instance in Jaako *et al.*, 'Ultrastrong-coupling phenomena beyond the Dicke model', PRA **94**, 033850 (2016). Therefore, geometric USC was reached before only with types of superconducting qubits that cannot compete in performance with the transmon qubit. This may be one reason why Refs. [12] and [13] only present spectroscopic studies of the USC regime.

Together with the recent approach in Ref. [16], we are the first to observe and investigate the dynamics of the quantum Rabi model in USC conditions, which is more feasible with a well performing qubit at hand.

While the slowing down of the system dynamics effectively reduces the qubit lifetime in the rotating frame, along with the cooperativity measure, the slowed-down dynamics allows to observe the quantum state collapse and revival signature on a much better experimentally accessible timescale. Since the revival rate is connected to the inverse bosonic mode frequency, a laboratory frame USC quantum Rabi Hamiltonian would exhibit revivals at a repetition rate on a sub-nanosecond scale. Conventional microwave equipment is generally limited to a sample rate of few gigahertz, which makes the system dynamics very challenging to trace. This advantage of the simulation is pointed out in the Discussion section of our manuscript.

The authors added a note in the final paragraph of the introduction, noting that the presented experiment is to be understood as a proof of principle for analog quantum simulation, rather than it revealing new physical insight to the quantum Rabi Hamiltonian.

- Although the manuscript by Braumüller et al. goes through the derivation of the effective quantum Rabi Hamiltonian in the main text and even more detail in the supplementary material, the idea and theory are entirely based on Ref. [25].

We agree with your remark. The authors however think that it is very beneficial for the readability and independence of the manuscript to provide the full theoretical basics of the simulation scheme. At the same

time, we reference the original theory proposal in the main text as well as in the Supplementary Material, pointing out explicitly that the presented scheme is based on the referenced proposal.

We present the calculation with a factor of two different in the applied drive amplitudes, since this appears more intuitive from an experimental point of view. The definition of the frequency constraint for switching on the second drive changes accordingly, such that a complete presentation of the modified calculation is useful.

In addition, we added some new calculations for the discussion of the parasitic driving, which requires to be put in mathematical context of the same notation.

- The experimental implementation of the two-tone drive and the creation of the effective Hamiltonian has already been demonstrated in Ref. [26], although for different parameters and with a different motivation.

The experiment in Ref. [26] indeed employs a drive scheme that is very similar to the one presented here, though with a different goal, as you pointed out.

A difference in the experiment by Vool *et al.* however is the use of one Rabi drive and one cavity drive, while the simulation scheme in our experiment requires two Rabi drives.

We point out the similarity to Ref. [26] in the manuscript in the section that introduces the simulation scheme.

In the light of the extensive previous comments from 3 referees in the first rounds of reviews, I will refrain from making more technical comments about the manuscript. However, I have been asked to comment on the reply to referee #3:

- When reading through the manuscript, I have found that the slowing down of the system dynamics relative to the coherence times of the system has been clearly introduced in this version.

- Referee #3 has made the valid point that the revival signature occurs at the same timescale as the system's coherence times. This makes it more difficult to observe the pure coherent dynamics, however as the authors point out, the collapse-revival signature is strong enough to originate convincingly from the USC physics.

- The authors have also added sufficient information about the future potential and how longer coherence times can be achieved in the experiment.

- Also the comment about the anharmonicity of the transmon being a limitation has been satisfactorily addressed.

Overall, I consider this manuscript a very strong and interesting piece of work. Although it might not contain any new or unexpected physics, it is still a proof-of-principle demonstration of the experimental feasibility of this simulation scheme. Hence, it will definitely be of high interest to specialists in the fields of superconducting cavity quantum electrodynamics and analog quantum simulation.